# Single-cell transcriptomics of human organoid-derived enteroendocrine cell populations from the small intestine

Christopher A. Smith , Van B. Lu , Rula Bany Bakar , Emily Miedzybrodzka , Adam Davison, Deborah Goldspink, Frank Reimann and Fiona M. Gribble

*Institute of Metabolic Science and MRC Metabolic Diseases Unit, University of Cambridge, Addenbrooke's Hospital, Cambridge, UK*

Handling Editors: Kim Barrett & Stephen Keely

The peer review history is available in the Supporting Information section of this article (https://doi.org/10.1113/JP287463#support-information-section).

**Abstract figure legend** Human intestinal organoids were genetically modified to express a fluorescent protein in enteroendocrine cells (EECs), which were then separated by flow cytometry and analysed by single-cell RNA sequencing. High numbers of human EECs were thereby available for cluster analysis and differential gene expression, revealing transcriptomic differences between EEC populations.

**Christopher A. Smith** is a Research Associate and Microscopy Facility Manager at the Institute of Metabolic Science at the University of Cambridge, UK. Chris is particularly interested in using bioinformatics approaches to investigate how enteroendocrine cells in the mammalian gut respond to various nutrients, and how the hormone peptides they secrete may interact with the brain. **Van B. Lu** is currently an Assistant Professor in the Department of Physiology & Pharmacology at the University of Western Ontario. Lu's research programme focuses on elucidating the nutrient-sensing mechanisms in gastrointestinal epithelial cells. This includes the role of bacterial metabolites in regulating cellular function and development, and gut barrier function.

C. A. Smith and V. B. Lu contributed equally to this work.
F. Reimann and F. M. Gribble are joint senior authors.

**Abstract** Gut hormones control intestinal function, metabolism and appetite, and have been harnessed therapeutically to treat type 2 diabetes and obesity. Our understanding of the enteroendocrine axis arises largely from animal studies, but intestinal organoid models make it possible to identify, genetically modify and purify human enteroendocrine cells (EECs). This study aimed to map human EECs using single-cell RNA sequencing. Organoids derived from human duodenum and ileum were genetically modified using CRISPR-Cas9 to express the fluorescent protein Venus driven by the chromogranin-A promoter. Fluorescent cells from CHGA-Venus organoids were purified by flow cytometry and analysed by 10X single-cell RNA sequencing. Cluster analysis separated EEC populations, allowing an examination of differentially expressed hormones, nutrient-sensing machinery, transcription factors and exocytotic machinery. Bile acid receptor *GPBAR1* was most highly expressed in L-cells (producing glucagon-like peptide 1 and peptide YY), long-chain fatty acid receptor *FFAR1* was highest in I-cells (cholecystokinin), K-cells (glucose-dependent insulinotropic polypeptide) and L-cells, short-chain fatty acid receptor *FFAR2* was highest in ileal L-cells and enterochromaffin cells, olfactory receptor *OR51E1* was notably expressed in ileal enterochromaffin cells, and the glucose-sensing sodium glucose cotransporter *SLC5A1* was highly and differentially expressed in K- and L-cells, reflecting their known responsiveness to ingested glucose. The organoid EEC atlas was merged with published data from human intestine and organoids, with good overlap between enteroendocrine datasets. Understanding the similarities and differences between human EEC types will facilitate the development of drugs targeting the enteroendocrine axis for the treatment of conditions such as diabetes, obesity and intestinal disorders.

(Received 29 August 2024; accepted after revision 12 November 2024; first published online 5 December 2024)

**Corresponding authors** F. Gribble and F. Reimann: Institute of Metabolic Science, Addenbrooke's Hospital, Cambridge, CB2 0QQ, UK. Email: fmg23@cam.ac.uk and fr222@cam.ac.uk

**Key points**

- Gut hormones regulate intestinal function, nutrient homeostasis and metabolism and form the basis of the new classes of drugs for obesity and diabetes.
- As enteroendocrine cells (EECs) comprise only ∼1% of the intestinal epithelium, they are under-represented in current single-cell atlases.
- To identify, compare and characterise human EECs we generated chromogranin-A labelled organoids from duodenal and ileal biopsies by CRISPR-Cas9.
- Fluorescent chromogranin-A positive EECs were purified and analysed by single-cell RNA sequencing, revealing predominant cell clusters producing different gut hormones.
- Cell clusters exhibited differential expression of nutrient-sensing machinery including bile acid receptors, long- and short-chain fatty acid receptors and glucose transporters.
- Organoid-derived EECs mapped well onto data from native intestinal cell populations, extending coverage of EECs.

## Introduction

Enteroendocrine cells (EECs) are the source of a variety of gut hormones that control local intestinal functions such as gallbladder contraction, pancreatic enzyme secretion and intestinal motility, as well as peripheral nutrient metabolism and appetite. They make up around 1% of all intestinal epithelial cells, and are constantly regenerated from stem cells in the crypts, with individual epithelial cells having a lifespan of around 5 days in the small intestine (Gribble & Reimann, 2019). Particularly relevant to the control of nutrient metabolism and food intake are the cells producing the hormones glucagon-like peptide-1 (GLP-1), which forms the basis for the new class of injectable diabetes and weight loss drugs (Drucker & Holst, 2023), cholecystokinin (CCK), peptideYY (PYY) and glucose-dependent insulinotropic polypeptide (GIP).

Recent years have witnessed an explosion of interest in, and understanding of, EEC function, made possible by the

use of transgenic mice and intestinal organoids expressing fluorescent markers driven by EEC-specific promoters, which enable live EEC identification and purification for transcriptomic and functional analysis (Beumer et al., 2020; Reimann et al., 2008). As a result of these studies, the traditional alphabetical EEC nomenclature in which e.g. I-cells produce CCK, K-cells produce GIP and L-cells produce GLP-1 and PYY, no longer represents our improved understanding of these cell populations, as several hormones traditionally ascribed to EECs with distinct single-letter codes actually arise from overlapping cell populations (Kaelberer & Bohorquez, 2018). Furthermore, studies on EEC nutrient-sensing mechanisms have revealed that different EEC sub-types express the same pool of receptors, including the sodium-coupled glucose cotransporter (SGLT1), and G-protein-coupled receptors (GPCRs) such as free fatty acid receptors (FFAR1–4), bile acid receptor (GPBAR1), 2-mono-oleoylglycerol receptor (GPR119), and amino acid receptors (CASR and GPR142) (Santos-Hernandez et al., 2024). Nevertheless, postprandial plasma excursions of different gut hormones are not identical (Veedfald et al., 2022), and likely reflect both the subtle characteristics of the cells producing them and their location along the gastrointestinal tract.

One method to generate an unbiased comparison between different EECs is single-cell RNA sequencing (scRNAseq), as the experimental design allows parallel analysis of large numbers of cells (e.g. ∼10,000 using 10X technology) from the same tissue preparation, collected and analysed in a single run. Although unbiased single-cell RNAseq analysis of the entire intestinal epithelium yields relatively few EECs due to their rarity, this can be improved using EEC-restricted fluorescent protein expression to enable EEC purification prior to analysis (Beumer et al., 2020). This method has been performed in transgenic mice using *NeuroD1* or *NeuroG3* as global EEC markers (Bai et al., 2022; Billing et al., 2019; Hayashi et al., 2023; Smith et al., 2024), or in human organoids in which EEC formation was enhanced by inducible *NEUROG3* expression (Beumer et al., 2020). In the current study, we generated fluorescent reporter organoids from the human duodenum and ileum using the chromogranin-A (CHGA) promoter as a pan-EEC marker (Beumer et al., 2020), and used scRNAseq to cluster and characterise the full EEC populations in the upper and lower small intestine. Although the expected nutrient-sensing receptors were spread across EEC clusters, differences between cell clusters were apparent using this approach. The dataset was merged with published data from native human intestine (Hickey et al., 2023) and organoids (Beumer et al., 2020), which contained fewer EECs, revealing good overlap between EEC clusters from the different studies.

## Methods

### Human organoid culture and generation of CHGA-Venus reporter lines

Human small intestinal (duodenum, ileum) organoids were generated from fully anonymised specimens provided by Addenbrooke's Hospital Tissue Bank with approval from the East of England – Cambridge Central Research Ethics Committee (ref: 09/H0308/24). Organoids were passaged every 10–14 days, maintained, and differentiated as previously described (Goldspink et al., 2020; Miedzybrodzka et al., 2021). The human organoid medium consisted of advanced DMEM:F12 (ADF) medium, conditioned Wnt3A and RSPO-1 medium, murine noggin (100 ng/ml, Peprotech), B27 supplement (Invitrogen), N2 supplement (Invitrogen), L-glutamine (2 mм, Sigma–Aldrich), penicillin-streptomycin (50 U/ml and 50 $\mu$g/ml, respectively, Sigma–Aldrich), *N*-acetyl-L-cysteine (1 mм, Sigma–Aldrich), murine epidermal growth factor (EGF, 50 ng/ml, Invitrogen), A83-01 (500 nм, Tocris), human [Leu15]-gastrin I (10 nм, Sigma–Aldrich), human insulin-like growth factor 1 (IGF-1, 100 ng/ml, Biolegend) and human fibroblast growth factor 2 (FGF-2, 50 ng/ml Peprotech). Y-27632 (10 $\mu$м, Tocris) was added to human organoid medium for 3–4 days after passaging. The differentiation medium consisted of human organoid medium without EGF.

Reporter lines were generated by homology-directed repair (HDR)-mediated CRISPR knock-in of the Venus gene and a neomycin-selection cassette. A CRISPR site on exon 6 of the chromogranin-A gene, which included the stop codon, was targeted with guides (ACCAGCTGCAGGCACTACGG and CCGTAGTGCCTGCAGCTGGT) cloned into plasmids co-expressing nuclear-targeted hSpCas9 (pX330, Addgene #42230). Homology donor regions were cloned into pTOPO plasmid (Invitrogen) by Gibson cloning. Plasmids (30 $\mu$g donor, 20 $\mu$g guide/Cas9 plasmid) were delivered to dissociated organoids by electroporation (Miedzybrodzka et al., 2021). Antibiotic selection with 0.5 mg/ml G418 began 4–6 days post-electroporation and surviving organoids were manually picked to establish clonal organoid lines. Integration was tested by PCR screening and confirmed by Sanger sequencing (Source BioScience).

### Fluorescence-activated cell sorting (FACS)

Human small intestinal organoids were cultured in a differentiation medium for 5–10 days before enzymatic and mechanical dissociation to single cells, as previously described (Goldspink et al., 2020; Miedzybrodzka et al., 2021). 16,000 and 13,124 Venus-positive live cells

(DAPI-negative, DRAQ5-positive) were collected from the human CHGA-labelled duodenal and ileal organoid lines, respectively, using a FACS Melody cell sorter (BD Biosciences, Cambridge Institute for Medical Research Flow Cytometry Core Facility).

### cDNA library and single-cell RNA sequencing

Collected cells from FACS were immediately processed for droplet encapsulation. cDNA libraries from purified Venus-positive cells were generated using the 10X Genomics Chromium platform and single-cell expression V3.1 reagents. Library sequencing was performed using an Illumina NovaSeq 6000 (Genomics and Transcriptomic Core, Institute of Metabolic Science).

Quality controls, read alignment (with reference to the GRCh38 genome), and raw count quantification for each cell were generated using the CellRanger pipeline (v6.1.2, 10X Genomics).

Analyses from raw counts were performed using the Seurat package (v5.0.1) in R (Butler et al., 2018). Samples were initially filtered per region so that each gene be detected in at least three cells, and each gene have at least 500 raw counts across all samples per region. Samples were further filtered so that the percentage of mitochondria-encoded genes (%mito) was less than 25% of all raw counts per sample. Raw count data were normalised using SCTransform, regressing %mito. Principal component analysis was performed, and the first 20 principal components were used to inform downstream Uniform Manifold Approximation and Projection (UMAP) and clustering analyses. Clustering of cells was performed by $k$-means, with $k = 12$ and resolution = 1.5.

The duodenum and ileum integrated datasets derived from individual datasets were integrated using Seurat's canonical correlation analysis-driven integration of SCTransform-normalised datasets. The integrated dataset was then re-clustered using $k$-means nearest neighbours, and each cluster was labelled based on genes found to be differentially expressed in each labelled cluster from each individual dataset. All EECs in the integrated dataset were separated from the other epithelial cell types and re-clustered again to better distinguish the individual EEC cell types. UMAPs in Fig. 4 are labelled as per the re-clustered data. The list of variable genes for integration was defined as the union of all variable genes per dataset, and the number of integration features used was 2000. The datasets used for integration were from previously published human duodenum and ileum epithelial cells (Becker, 2023; Hickey et al., 2023), and human duodenum and ileum organoids (Beumer et al., 2020), extracted from the Gene Expression Omnibus (GEO; GSE146799). Data from published datasets were normalised using SCTransform within Seurat prior to integration. Integration was performed as above.

## Results

CHGA-Venus labelled human duodenal and ileal organoids were generated using CRISPR-Cas9-mediated HDR using the strategy shown in Fig. 1*A*. Organoids from both regions exhibited scattered fluorescent cells with morphology typical of EECs (Fig. 1*B* and *C*). Fluorescent CHGA-positive cells from both intestinal regions were collected by flow cytometry for 10X droplet-based library preparation and single-cell RNA sequencing (Fig. 1*D* and *E*).

Analysis revealed higher expression of particular hormones in distinct clusters corresponding to the established one-letter code for EEC naming (Fig. 2*A* and *B*), even though most hormones were detectable in several clusters (Fig. 2*C–E*). In duodenal CHGA-Venus organoids, the prominent EEC types were enterochromaffin cells (EC cells, characterised by *TPH1* expression), K-cells (expressing *GIP*), I-cells (expressing *CCK*), M/X cells (expressing *MLN* and *GHRL*) and D-cells (expressing *SST*). In ileal CHGA organoids, the major EEC clusters were EC cells, L-cells, M/X cells and D-cells. A subset of cells from the duodenum expressed a number of hormones at lower levels than other clusters and lacked a hormone expressed at higher levels – these are denoted as polyhormonal cells in the figures. Expression of some gut hormones showed overlap between certain EEC populations, such as the detection of *CCK* in K-cells, consistent with previous reports from the mouse intestine (Bai et al., 2022; Habib et al., 2012; Hayashi et al., 2023; Smith et al., 2024). The duodenal I-cell and K-cell populations co-expressed *GAST*, as also previously identified in human NEUROG3-driven human duodenal organoids (Beumer et al., 2020). *SCT* expression was very infrequent, likely reflecting a lack of the most mature population of EECs in these organoids. In addition to the hormones defining the cell clusters, human EECs also expressed low levels of *TAC3, NPW* and *VGF*, with no apparent predominance in any EEC cluster. *TAC1* expression was lower than expected for crypt-like ECs based on data from mice (Beumer et al., 2020; Hayashi et al., 2023; Roth & Gordon, 1990), and was largely restricted to ileal ECs. Amongst the chromogranin/secretogranin family, *CHGA* and *SCG5* exhibited higher expression in EECs from the ileum, and *SCG2* was more highly expressed in duodenum (Fig. 2*E*).

We next sought to evaluate the expression of GPCRs and nutrient transporters potentially involved in detecting food ingestion (Fig. 3*A*). *FFAR1* expression was highest in I-cells and L-cells; *GPBAR1* was highest in ileal L-cells but also detected in most other EEC populations; *FFAR4* was highest in M/X cells and ileal L-cells; *GPR119* in L-cells; *OR51E1* in a number of EEC types and notably higher in cells from ileum than duodenum. Compared with other EEC types, EC cells expressed low levels of *FFAR1/4*,

*GPR119* and *CASR*. The intestinal glucose sensor SGLT1 (encoded by *SLC5A1*), was notably most highly expressed in L-cells and K-cells.

Figure 3*B* shows expression of GPCRs potentially involved in paracrine cross-talk between different EEC populations. Notably, *GIPR* was expressed across all human EEC cell populations; *GLP1R* expression was less common, and mostly restricted to EECs from the duodenum. *RXFP4*, the receptor for INSL5, a hormone co-released by colonic L-cells (Billing et al., 2019), was broadly expressed in EC cells, and was also detectable in scattered EECs from other clusters in the duodenum. *NPY1R* was predominantly located in ileal EC cells, suggesting that there may be inhibitory paracrine cross-talk between PYY and 5-HT release (as NPY1R is Gi coupled). Amongst the SST receptor family, *SSTR2* was most highly expressed, with little expression of *SSTR5*, contrasting with functional data in rodents showing strong activity of SSTR5 in intestinal EECs (Jepsen et al., 2021; Moss et al., 2012).

Transcription factors relevant to EEC specification in the human intestinal epithelium were examined (Fig. 3*C*). This confirmed the previously reported selective enrichment of *LMX1A* and (to a lesser extent) *MNX1* in ECs (Beumer et al., 2020). *HHEX* was highly localised to *SST*-expressing D-cells, while the relative absence of *ARX*

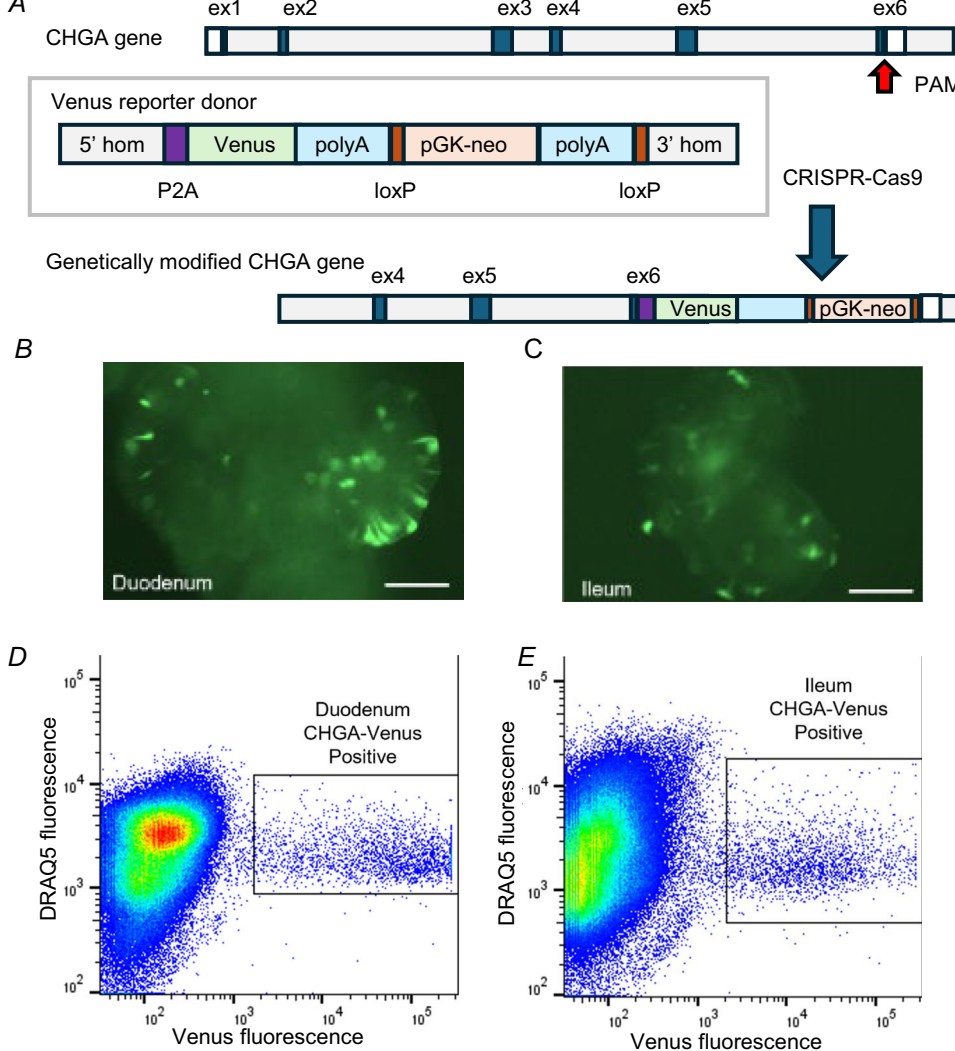

**Figure 1. Generation of, and purification of enteroendocrine cells (EECs) from, chromogranin-A (CHGA)-Venus human organoids**
*A*, schematic of the cloning strategy used to insert the Venus transgene into exon 6 of the CHGA locus. *B* and *C*, live images of CHGA-Venus human organoids from duodenum (*B*) and ileum (*C*). Scale bar 400 $\mu$m. *D* and *E*, fluorescence-activated cell sort plots of 500 k events from CHGA-Venus duodenal (*D*) and ileal (*E*) organoids. CHGA-Venus positive and negative cells were isolated based on Venus fluorescence intensity, after the selection of live DAPI-negative and DRAQ5-positive cells only.

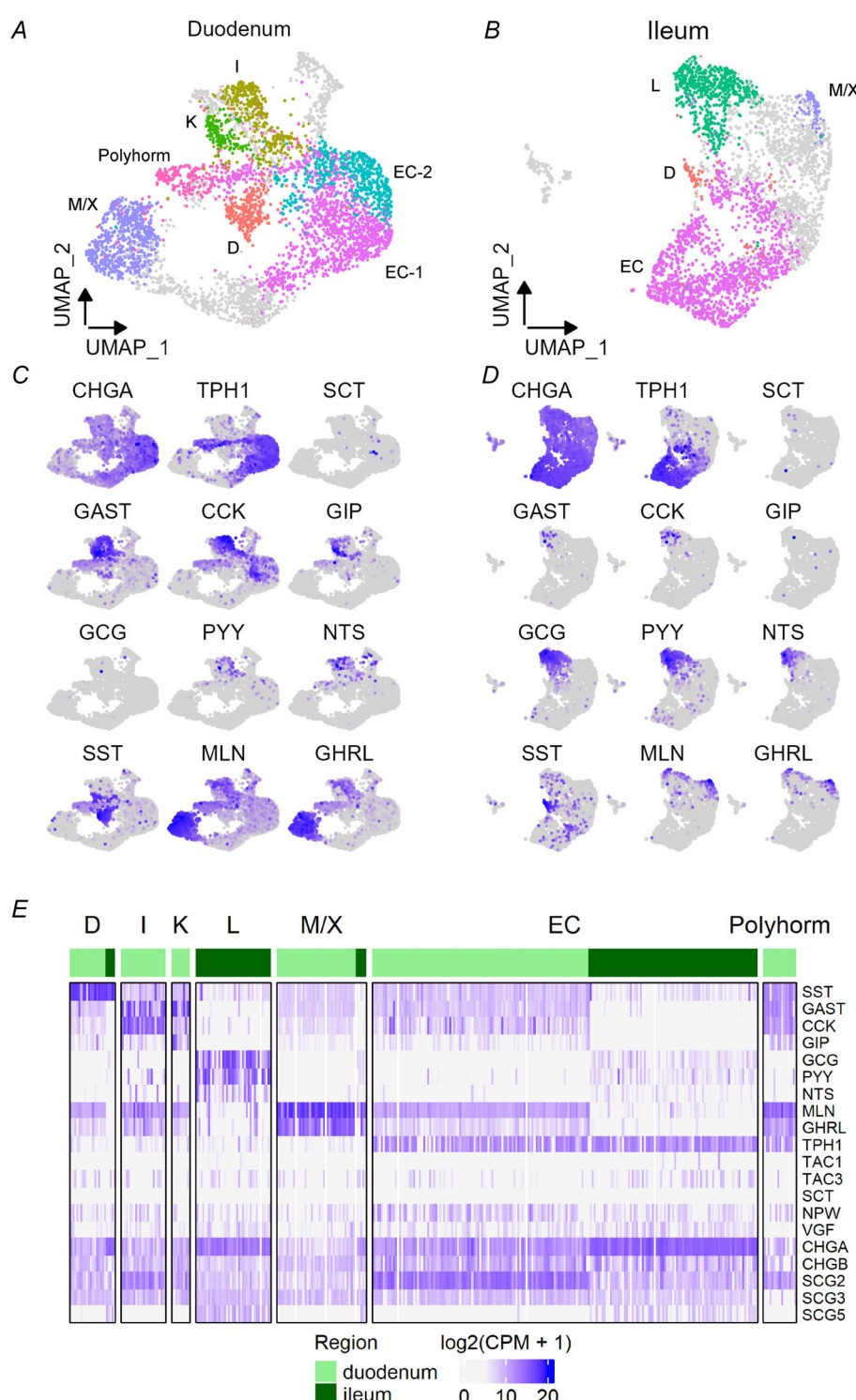

**Figure 2. Clustering and annotation of chromogranin-A (CHGA)-Venus 10X scRNAseq data**

*A* and *B*, clustering and cluster-annotation of single-cell RNA sequencing data performed on duodenal and ileal enteroendocrine cells (EECs) fluorescence-activated cell sort purified from CHGA-Venus human organoids. Dimension reduction was performed using Uniform Manifold Approximation and Projection. Clusters were identified and defined by the feature maps of marker genes shown in (*C* and *D*) (*EC, TPH1*; *D, SST*; *I, CCK*; *K, GIP*; *L, GCG*; *M, MLN*; *X, GHRL*). Polyhorm denotes a population of cells expressing low levels of multiple hormones. *E* and *F*, heatmaps showing expression of hormones, EEC markers and members of the chromogranin family in CHGA-Venus cells, arranged by cell type and intestinal region. CPM, counts per million.

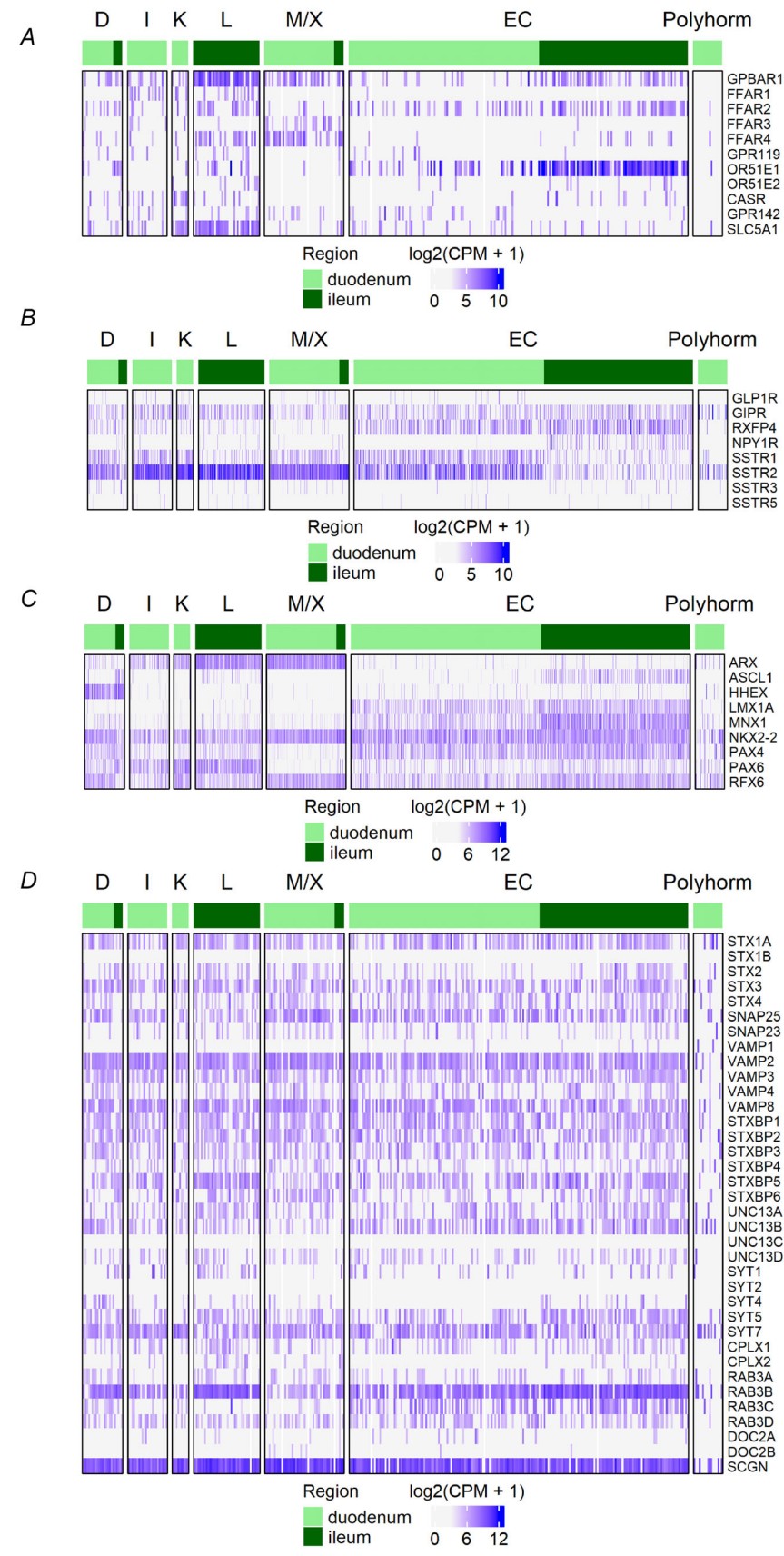

**Figure 3. Expression of genes involved in enteroendocrine cell (EEC) development and function**

Heatmaps showing expression in chromogranin-A (CHGA)-Venus derived EECs of (A) receptors involved in sensing nutrient ingestion and microbial metabolites, (B) receptors potentially involved in paracrine cross-talk between EECs, (C) transcription factors known to play a role in EEC development and (D) exocytosis-related genes. Data are arranged by cell type and intestinal region. CPM, counts per million.

in ECs and D-cells was also noted. *PAX6* was most highly expressed in K- and L-cells from duodenum and ileum, respectively.

As EECs release their hormonal contents by regulated exocytosis, we investigated the expression of machinery potentially involved in vesicle function and release. The secretory machinery itself was largely similar across EEC clusters, although with some minor variation between clusters and regions (Fig. 3*D*). All EEC populations expressed the core SNARE complex proteins (*SNAP25*, *STX1A* and *VAMP2*), accessory proteins implicated in vesicular docking and priming (*STXBP1*, *UNC13A/B*, *RAB3B*, *CPLX1/2*) and the calcium-binding protein secretogogin (*SCGN*). We note, however, that the cluster of polyhormonal cells had fewer hits for secretory machinery genes, suggesting this may represent immature EECs not yet committed to a definitive hormone.

Data from CHGA-Venus organoids were combined with publicly available single-cell RNA sequencing data from the native human intestine (Hickey et al., 2023) and organoids with NEUROG3-driven enhanced secretory line differentiation (Beumer et al., 2020). Overlaps between the EEC populations from the three datasets were good for both duodenum (Fig. 4*A–C*) and ileum (Fig. 4*D* and *E*), supporting the validity of studying EECs derived from organoids. The new data from CHGA-Venus organoids add a substantial number of EECs to this combined cluster map.

## Discussion

By analysis of fluorescently labelled EECs purified from CHGA-Venus organoids, the current study provides single-cell expression data on organoid-derived EECs from the human duodenum and ileum, adding to the previously published relatively small EEC dataset from human organoids driven by induced *NEUROG3* over-expression (Beumer et al., 2020), and EEC populations from native human small intestine (Beumer et al., 2020; Hickey et al., 2023). Importantly, merging the datasets from these three studies revealed strong over-

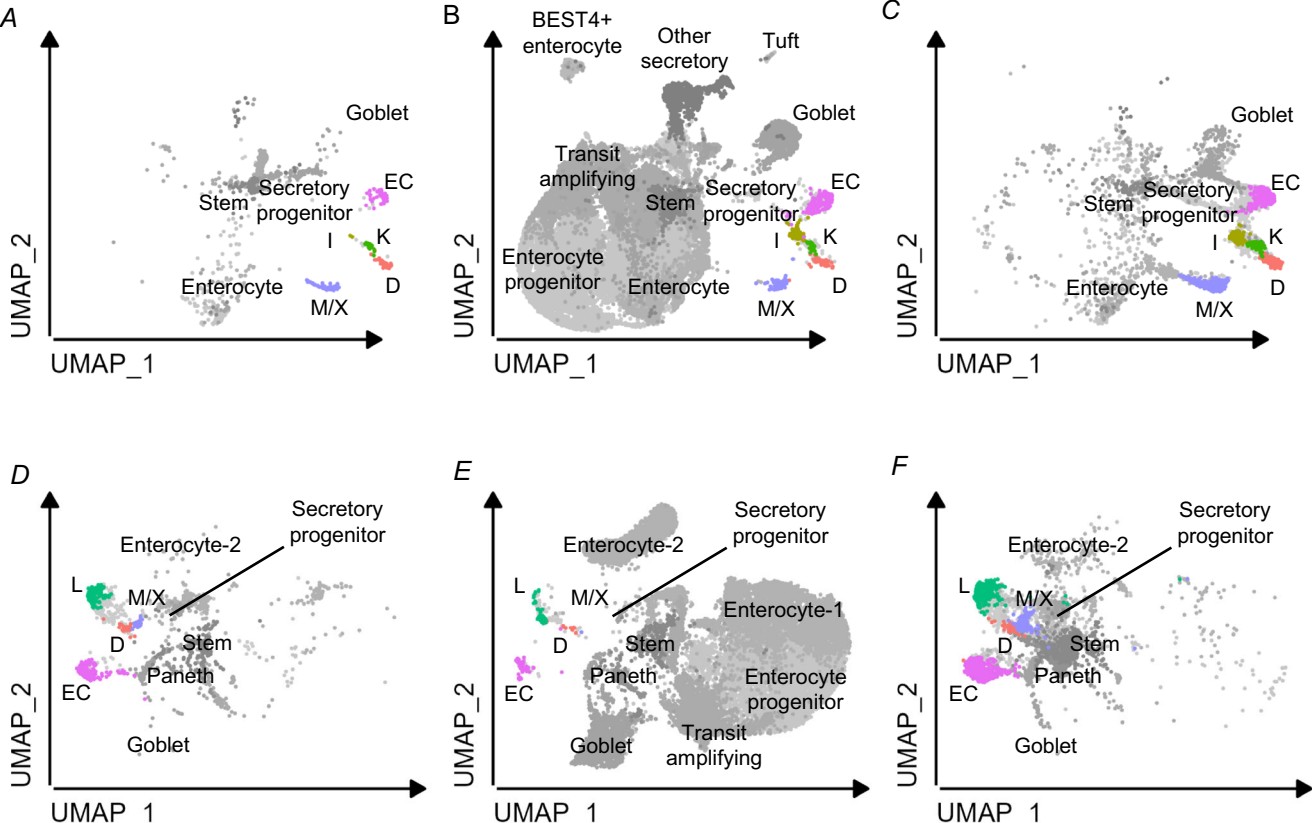

**Figure 4. Merging data from chromogranin-A (CHGA)-Venus organoids with published human datasets**
Published single-cell RNA sequencing data from human intestinal organoids (Beumer et al., 2020) and native human intestine (Hickey et al., 2023) were merged with the new data from CHGA-Venus organoids, to generate a combined Uniform Manifold Approximation and Projection plot. Contributions of cells from the different studies are depicted separately in the different panels: *A* and *D* show the data from Beumer et al. (2020); *B* and *E* show the data from Hickey et al. (2023), including associated cell annotations; and *C* and *F* show the new data from CHGA-Venus organoids. Colours depict the different EEC clusters.

lap of the EEC clusters. Cluster analysis of CHGA-Venus organoid-derived EECs identified the following major cell populations (in descending order of cluster size): duodenum – EC > M/X > I > D > K-cells; ileum – EC > L > M/X > D-cells. Some hormone genes were detected at low levels in the majority of cells isolated from that intestinal region, and could reflect mRNA contamination of the sheath fluid arising from lysis of a few cells expressing very high levels of the hormone during the microfluidic cell separation. While this could contribute to the appearance of the polyhormonal cell cluster, the alternative explanation that some EECs co-express a number of hormones is supported by immunohistochemical and mass-spectrometric studies, showing, for example, that CCK is produced by murine K- and L-cells (Cho et al., 2015; Habib et al., 2012) and human K- and M/X cells (Guccio et al., 2024; Miedzybrodzka et al., 2021). A distinct duodenal I-cell cluster was identified that had not been found in the previously published human organoid study (Beumer et al., 2020) but was evident in native duodenum (Hickey et al., 2023), allowing transcriptomic comparisons to be made with this important CCK-producing cell type. *SCT* expression was largely undetectable in organoid-derived EECs, as also reported previously (Beumer et al., 2020), although a distinct population of S-cells was reported in the native human EEC dataset (Hickey et al., 2023). As *SCT* is predominantly expressed in mature EECs in duodenal villi (Aiken & Roth, 1992; Hysenaj et al., 2024), its relative absence in the organoid-derided cell populations suggests that the organoids are biased towards generation of crypt over villus EECs.

Relatively little is understood about how transcription factors drive formation of the different EEC clusters, but a few clear patterns were observable in our TF analysis: (1) *HHEX* was largely restricted to D-cells, as also previously observed in mouse and human intestinal organoids, and mirroring the restricted expression of *Hhex* in other somatostatin-expressing cells such as pancreatic delta cells (Zhang et al., 2014) and gastric D-cells (Adriaenssens et al., 2015); (2) *ARX* was particularly highly expressed in L, K, M/X and I-cells; which set of genes it regulates in EECs is unclear but it has previously been implicated in endocrine cell specification in the endocrine pancreas (Gage et al., 2015); (3) *LMX1A* and *MNX1* were most highly expressed in EC cells, as reported previously (Beumer et al., 2020); and (4) the TF expression profiles of K, I and L-cells were largely indistinguishable, consistent with the strong overlaps between these cell populations found in a number of studies (Egerod et al., 2012; Haber et al., 2017; Habib et al., 2012; Kaelberer & Bohorquez, 2018; Smith et al., 2024).

Transcriptomic characteristics of human EEC clusters were broadly similar to those described previously in

mice, isolated from intestinal tissue labelled by fluorescent reporter expression controlled by NeuroD1-Cre expression (Smith et al., 2024) but with several notable differences. Whereas a number of analyses of mouse EECs have revealed strong overlap between L-cells and I-cells (Egerod et al., 2012; Habib et al., 2012; Smith et al., 2024), these clusters were more distinct in the human data, with relatively low co-expression of *GCG* and *PYY* in duodenal 'I-cells', or of *CCK* in ileal 'L-cells'. Instead, the human I-cell cluster exhibited relatively high expression of *GAST*. *GIP* was similarly co-localised with *GAST*, as also reported previously in human organoids (Beumer et al., 2020). Production of GAST in the human small intestine does not seem to be an artefact of generating EECs in organoids, as our previous studies identified relatively high levels of GAST peptide in human duodenal extracts and of GAST mRNA and protein in purified native jejunal EECs (Roberts et al., 2019). In mouse, by contrast, there are very few *Gast*-positive cells outside the stomach and no evident co-localisation with *Gip* (Smith et al., 2024). Alongside this study, we further optimised differentiation protocols to generate mature functional K-cells in human GIP-Venus organoids (Guccio et al., 2024), which also expressed high levels of GAST mRNA and peptide; however, the physiological role of duodenal-derived gastrin is currently obscure.

Although nutrient-sensing GPCRs have been identified in many different EEC populations (mouse and human) by bulk RNA sequencing, there is less understanding of how expression levels vary between EEC cell types. Applying scRNAseq to CHGA-labelled cells enabled direct comparisons between EEC clusters and revealed several notable differences in this study. *SLC5A1* expression was highest in K- and L-cells, consistent with the extensively documented ability of oral glucose to trigger elevations in plasma GIP and GLP-1 (Nauck et al., 1993), attributed mechanistically to the action of SGLT1 as the intestinal glucose sensor (Gorboulev et al., 2012). *GPBAR1* was identified in all EEC clusters but was particularly highly expressed in L-cells and was higher in the ileum than duodenum, consistent with the ileum being the major site of bile acid reabsorption – a step previously shown to be essential for bile acids to reach *GPBAR1* located on the basolateral surface of L-cells (Brighton et al., 2015). The long-chain free fatty acid receptor *FFAR1* was most highly expressed in K-, I- and L-cells, mirroring the well-documented FFAR1-dependent increases in plasma GIP, CCK and GLP-1 levels following lipid ingestion (Hauge et al., 2015; Liou et al., 2011). *FFAR4* expression was highest in M/X cells: although FFAR4 has been linked to gut hormone secretion in a number of studies, the function of FFAR4 in EECs remains uncertain (Hirasawa et al., 2005; Sankoda et al., 2017) and we were previously unable

to demonstrate any stimulatory contribution of FFAR4 agonists to MLN secretion in human MLN-labelled organoids (Miedzybrodzka et al., 2021). Notable amongst the receptors for short- and branched-chain fatty acids were the high expression of *FFAR2* in L-cells and of both *FFAR2* and *OR51E1* in ileal ECs, suggesting that small intestinal microbial fermentation products might exert their strongest effects on the secretion of GLP-1, PYY and 5-HT. Roles for Ffar2 and the OR51E1 homologue Olfr558 have been demonstrated previously in murine L-cells and EC cells (Bellono et al., 2017; Tolhurst et al., 2012). Subtle differences in the nutrient-sensing machinery between some EEC clusters likely contribute to different patterns of plasma gut hormone concentrations measured after food ingestion.

Hormone secretion by EECs occurs through calcium-dependent vesicular exocytosis, a highly regulated process involving a cohort of conserved core SNARE complex proteins alongside a range of accessory and regulatory proteins, the composition and function of which have yet to be characterised in human EECs (Sudhof, 2013). We found that, similar to most neuronal and neuroendocrine cell types which exhibit regulated calcium-dependent exocytosis, all populations of human ileal and duodenal EECs identified expressed the core SNARE complex proteins (*SNAP25*, *STX1A* and *VAMP2*) and accessory proteins implicated in vesicular docking and priming (*STXBP1*, *UNC13A/B*, *RAB3B* and *CPLX1/2*). This suggests that exocytosis in EECs likely occurs through mechanisms similar to those of other secretory cells, although this is yet to be confirmed with functional studies. We also found that all EECs showed expression of exocytotic proteins known to be particularly enriched in neuroendocrine cells such as those found in the pancreatic islets, specifically the calcium-binding proteins *SYT7* and *SCGN* (Gustavsson et al., 2009; Wagner et al., 2000). Expression of many of the exocytotic proteins in EECs appears to be consistent between human and mouse, where the core SNARE complex proteins STX1A, SNAP25 and VAMP2, and the accessory protein STXBP1 have been demonstrated to have functional roles in exocytosis in mouse L-cells (Campbell et al., 2020; Li et al., 2014; Wheeler et al., 2017). Expression of exocytotic proteins was broadly similar across all EECs meaning that the terminal steps of secretion are likely regulated in a similar manner among the different EEC types. However, we did observe an enrichment of *SYT1* in the ileal population of EECs, the calcium sensor responsible for fast and synchronous vesicular exocytosis in neuronal chemical synapses (Sudhof, 2013). This is interesting given that there is some evidence that EECs may be capable of fast neurotransmitter-mediated transmission to neuronal afferents (Kaelberer et al., 2018; Lu et al., 2019).

## Conclusions

Following the success of gut hormone-based therapies for treating intestinal disorders, diabetes and obesity, EECs are a prime target for future drug development. Despite the widespread application of single-cell transcriptomics to generate atlases of cell types in the human body, there are insufficient EECs in most databases to enable robust cluster analysis because the entire EEC population only makes up around 1% of the intestinal epithelium. By performing scRNAseq on EECs purified from CHGA-Venus organoids, we have substantially increased the number of EECs for analysis. Human EECs generated in organoids replicate the transcriptomic properties of their native counterparts, making them a good model for studying EEC physiology. Supporting the relevance of the transcriptomic approach taken here, a number of studies have demonstrated that EECs in human organoids replicate physiological properties of the native enteroendocrine system, employing nutrient-sensing machinery such as SGLT1, FFAR1 and GPBAR1 (Goldspink et al., 2020; Guccio et al., 2024; Miedzybrodzka et al., 2021). The ability to knock out candidate genes in organoids using CRISPR-Cas9 further offers a viable and valuable approach to validate the physiological relevance of individual genes identified in transcriptomic studies, complementing more traditional pharmacological approaches that are frequently hampered by limitations in drug specificity. The lack of *SCT*-expressing cells in organoids generated by current protocols suggests they have not reached full maturity, and while they are a good *in vitro* model system for the human enteroendocrine system, there is still a place for studies on intact perfused intestinal models, free-living animals and human volunteers where the mucosa is mature, intact and in contact with other cell types not present in the organoid system such as inflammatory cells and neurones.

Although the data clustered broadly into cell types according to the dominantly expressed hormone, mirroring the classical single-letter nomenclature, overlaps of hormone expression were apparent and are supported by many previous studies (Egerod et al., 2012; Haber et al., 2017; Habib et al., 2012; Kaelberer & Bohorquez, 2018) raising the question of whether it is time for the field to agree a new nomenclature for EECs that is both intuitive and representative.

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

## Additional information

### Data availability statement

CHGA-Venus duodenum and ileum processed Seurat objects, and raw data, will be made available on the Gene Expression Omnibus (GEO). Reference will be provided upon download.

### Competing interests

F.M.G. and F.R. have received funding from AstraZeneca and Eli Lilly for projects not overlapping the material in this manuscript. They also received sponsorship to host the European Incretin Study Group meeting in Cambridge 2024 from AstraZeneca, Eli Lilly, Sun Pharma and Mercodia. D.G. is currently an employee of GlaxoSmithKline, but her work towards this manuscript was performed when an employee of the University of Cambridge and is unrelated to and independent of her position at GSK.

### Author contributions

C.S., F.R. and F.G.: Conception or design of the work; acquisition, analysis or interpretation of data for the work; drafting the work or revising it critically for important intellectual content; final approval of the version to be published; and agreement to be accountable for all aspects of the work. V.L., R.B.B., E.M., A.D. and D.G.: Conception or design of the work; acquisition, analysis or interpretation of data for the work; drafting the work or revising it critically for important intellectual content; and final approval of the version to be published. All authors have approved the final version of the manuscript and agree to be accountable for all aspects of the work. All persons designated as authors qualify for authorship, and all those who qualify for authorship are listed.

### Funding

This research was funded by a Wellcome joint investigator award to FR/FMG (220271/Z/20/Z) and the MRC Metabolic Diseases Unit (MRC_MC_UU_12012/3). Core support was provided by MRC [MRC_MC_UU_00014/5] and Wellcome [100574/Z/12/Z]). F.G. and F.R. received funding from AstraZeneca and Eli Lilly for non-overlapping research on other projects. They received sponsorship from AstraZeneca, Eli Lilly, Sun Pharma and Mercodia to run the 5th European Incretin Study Group conference in Cambridge (April 2024).

### Acknowledgements

We thank the MRL Genomics and Transcriptomics Core, the Core Biochemical Assay Laboratory (CBAL), the Flow Cytometry Core at CIMR, CRUK Cambridge Institute Genomics Core and Addenbrooke's Tissue Bank.

### Author's present addresses

V.B. Lu: Department of Physiology and Pharmacology, University of Western Ontario, London, ON, Canada N6A 5C1. R. Bany Bakar: Department of Medicine, University of Cambridge, Addenbrooke's Hospital, Cambridge, UK. E.Miedzybrodzka: Centre for Regenerative Medicine, Institute for Regeneration and Repair, The University of Edinburgh, 5 Little France Drive, Edinburgh EH16 4UU, UK. D.Goldspink: GlaxoSmithKline, Medicines Research Centre, Gunnels Wood Road, Stevenage, SG1 2NY, UK.

### Keywords

chromogranin, enteroendocrine, GLP-1, GIP, gut hormone, organoid, single cell RNA sequencing

## Supporting information

Additional supporting information can be found online in the Supporting Information section at the end of the HTML view of the article. Supporting information files available:

**Peer Review History**

