## [Peer Review History · The Journal of Physiology]

Single cell transcriptomics of human organoid-derived enteroendocrine cell populations from the small intestine

Christopher A. Smith, Van B Lu, Rula Bany Bakar, Emily Miedzybrodzka, Adam Davison, Deborah Goldspink, Frank Reimann, and Fiona Gribble

DOI: 10.1113/JP287463

Corresponding author(s): Fiona Gribble (fmg23@cam.ac.uk)

The following individual(s) involved in review of this submission have agreed to reveal their identity: Carel Le Roux (Referee #1)

Review Timeline:

Submission Date:	29-Aug-2024
Editorial Decision:	27-Sep-2024
Revision Received:	09-Oct-2024
Editorial Decision:	15-Oct-2024
Revision Received:	23-Oct-2024
Accepted:	12-Nov-2024

Senior Editor: Kim Barrett

Reviewing Editor: Stephen Keely

Transaction Report:

Dear Dr Gribble,

Re: JP-RP-2024-287463 "Single cell transcriptomics of human organoid-derived enteroendocrine cell populations from the small intestine" by Christopher A. Smith, Van B Lu, Rula Bany Bakar, Emily Miedzybrodzka, Adam Davison, Deborah Goldspink, Frank Reimann, and Fiona Gribble

Thank you for submitting your manuscript to The Journal of Physiology. It has been assessed by a Reviewing Editor and by 2 expert referees and we are pleased to tell you that it is acceptable for publication following satisfactory revision.

REVISION CHECKLIST:

Please upload two versions of your manuscript text: one with all relevant changes highlighted and one clean version with no changes tracked. The manuscript file should include all tables and figure legends, but each figure/graph should be uploaded as separate, high-resolution files. The journal is now integrated with Wiley's Image Checking service. For further details, see: <https://www.wiley.com/en-us/network/publishing/research-publishing/trending-stories/upholding-image-integrity-wileys->

image-screening-service

We look forward to receiving your revised submission.

Yours sincerely,

Kim Barrett
Senior Editor
The Journal of Physiology

REQUIRED ITEMS

- Author photo and profile. First or joint first authors are asked to provide a short biography (no more than 100 words for one author or 150 words in total for joint first authors) and a portrait photograph. These should be uploaded and clearly labelled together in a Word document with the revised version of the manuscript. See Information for Authors for further details.

- Your manuscript must include a complete Additional Information section, including competing interests; funding; author contributions and acknowledgements.

- Please include an Abstract Figure file, as well as the Figure Legend text within the main article file. The Abstract Figure is a piece of artwork designed to give readers an immediate understanding of the research and should summarise the main conclusions. If possible, the image should be easily 'readable' from left to right or top to bottom. It should show the physiological relevance of the manuscript so readers can assess the importance and content of its findings. Abstract Figures should not merely recapitulate other figures in the manuscript. Please try to keep the diagram as simple as possible and without superfluous information that may distract from the main conclusion(s). Abstract Figures must be provided by authors no later than the revised manuscript stage and should be uploaded as a separate file during online submission labelled as File Type 'Abstract Figure'. Please also ensure that you include the figure legend in the main article file. All Abstract Figures should be created using BioRender. Authors should use The Journal's premium BioRender account to export high-resolution images. Details on how to use and access the premium account are included as part of this email.

EDITOR COMMENTS

Reviewing Editor:

Both Reviewers found significant merit and novelty in these studies and suggest only minor revisions, including some commentary in the Discussion regarding the "outdated" nomenclature used for EECs, ii) the utility for organoids in replacing or complementing studies more established animal models, and iii) the limitations of extrapolating transcriptome data alone to physiological function. I agree that addressing these concerns will strengthen the manuscript.

REFEREE COMMENTS

Referee #1:

The authors suggested that intestinal organoid models make it possible to identify, genetically modify and purify human enteroendocrine cells. They aimed to map human enteroendocrine cells using single-cell RNA-sequencing. Organoids derived from human duodenum and ileum were genetically modified and enteroendocrine cells purified by flow cytometry and

analysed by single-cell RNA-sequencing. Cluster analysis separated enteroendocrine cell populations, allowing examination of differentially-expressed hormones, nutrient sensing machinery, transcription factors and exocytotic machinery. They found that bile acid receptors was highly expressed in L-cells (producing glucagon-like peptide 1 and peptide YY), long chain fatty acid receptor FFAR1 was highest in I-cells (cholecystokinin), K-cells (glucose-dependent insulinotropic polypeptide) and L-cells, short chain fatty acid receptor FFAR2 was highest in ileal L-cells and enterochromaffin cells, olfactory receptor OR51E1 was notably expressed in ileal enterochromaffin cells, and the glucose-sensing sodium glucose cotransporter SLC5A1 was highly and differentially-expressed in K- and L-cells. The authors then merged the organoid enteroendocrine cell atlas with published data from human intestine and organoids. The authors found good overlap between enteroendocrine datasets.

This is a very good paper from a very good lab. The work is clearly describes and makes an important scientific contribution.

Comment 1. The authors make a very good case that the historical naming of enteroendocrine cells as D, I, K, L are no longer fit for purpose but then they revert back to the historical nomenclature. Can the authors in their limitation section explain why they revert back to a nomenclature?

Comment 2. Can the authors discuss in more detail whether organoids are able to substitute for older techniques such as those used in animal studies or whether there is a still a place for the older techniques.

Referee #2:

Title: Single cell transcriptomics of human organoid-derived enteroendocrine cell populations from the small intestine

The authors have formulated a transcriptomic analysis of ileal and duodenal enteroendocrine cells (EECs) purified from human intestinal organoids on the basis of Venus fluorescent protein reporter expression downstream of the Chromogranin A promoter. EEC clusters at the transcript level are described according to hormone synthesis, and expression of nutrient-sensing and exocytotic machinery, and transcription factors. Expression patterns are mapped to a similar high impact study performed on NEUROG3 derived human EECs from organoids (Beumer et al., 2020) and to ECs derived from the native intestine (Beumer et al., 2020; Hickey et al., 2023), where there is a high degree of congruence, and some notable differences. These findings are discussed in a sophisticated fashion.

There are two main limitations to the ms in it's current form: (i) transcriptomic expression patterns are not corroborated by immunolabelling of EEC derived from the organoids; I think this would be an important step towards validating the organoids as a valuable model for functional analysis; (ii) functional analysis of exemplar 'nutrient-sensing'-'hormone secretion' transcriptional expression patterns identified in EEC-subtypes. These gaps in the current study of organoids are partially addressed by referring to published work. It is fair to say that the title states that the study is a transcriptional analysis. Nevertheless, these limitations should be discussed in the context of future directions. Comments on expected findings and their implications would be informative. Also, implications of the current findings for human health and disease would also be informative and provide potential impact.

END OF COMMENTS

EDITOR COMMENTS

Reviewing Editor:

Both Reviewers found significant merit and novelty in these studies and suggest only minor revisions, including some commentary in the Discussion regarding (i) the "outdated" nomenclature used for EECs, ii) the utility for organoids in replacing or complimenting studies more established animal models, and iii) the limitations of extrapolating transcriptome data alone to physiological function. I agree that addressing these concerns will strengthen the manuscript.

Response: We thank the editor and reviewers for their positive comments and have made revisions as requested and as detailed below. Revisions are highlighted in the revised version.

REFEREE COMMENTS

Referee #1:

The authors suggested that intestinal organoid models make it possible to identify, genetically modify and purify human enteroendocrine cells. They aimed to map human enteroendocrine cells using single-cell RNA-sequencing. Organoids derived from human duodenum and ileum were genetically modified and enteroendocrine cells purified by flow cytometry and analysed by single-cell RNA-sequencing. Cluster analysis separated enteroendocrine cell populations, allowing examination of differentially-expressed hormones, nutrient sensing machinery, transcription factors and exocytotic machinery. They found that bile acid receptors was highly expressed in L-cells (producing glucagon-like peptide 1 and peptide YY), long chain fatty acid receptor FFAR1 was highest in I-cells (cholecystokinin), K-cells (glucose-dependent insulinotropic polypeptide) and L-cells, short chain fatty acid receptor FFAR2 was highest in ileal L-cells and enterochromaffin cells, olfactory receptor OR51E1 was notably expressed in ileal enterochromaffin cells, and the glucose-sensing sodium glucose cotransporter SLC5A1 was highly and differentially-expressed in K- and L-cells. The authors then merged the organoid enteroendocrine cell atlas with published data from human intestine and organoids. The authors found good overlap between enteroendocrine datasets.

This is a very good paper from a very good lab. The work is clearly describes and makes an important scientific contribution.

Comment 1. The authors make a very good case that the historical naming of enteroendocrine cells as D, I, K, L are no longer fit for purpose but then they revert back to the historical nomenclature. Can the authors in their limitation section explain why they revert back to a nomenclature?

Response: We were ourselves surprised that the data separated into clusters that broadly aligned with the classical single letter nomenclature, with individual clusters dominated by a given hormone. For this reason and to compare easily with the existing literature we kept the old single letter classification. However, many cells express more than one hormone and overlaps of different combinations are apparent in each cluster. This raises an interesting question of whether we should hold a symposium targeting this very question, and then write a position paper. We think it would require the support of a range of key stakeholders in the field if any new naming system were to be universally adopted. We have added the following text to the discussion:

“Although the data clustered broadly into cell types according to the dominantly expressed hormone, mirroring the classical single letter nomenclature, overlaps of hormone expression were apparent and are supported by many previous studies (Egerod *et al.*, 2012; Habib *et al.*, 2012; Haber *et al.*, 2017; Kaelberer & Bohorquez, 2018) raising the question of whether it is time for the field to agree a new nomenclature for enteroendocrine cells that is both intuitive and representative. .”

Comment 2. Can the authors discuss in more detail whether organoids are able to substitute for older techniques such as those used in animal studies or whether there is still a place for the older techniques.

Response: We have added some additional discussion which reads:

“The lack of *SCT*-expressing cells in organoids generated by current protocols suggests they have not reached full maturity, and while they are a good in vitro model system for the human enteroendocrine system, there is still a place for studies on intact perfused intestinal models, free-living animals and human volunteers where the mucosa is mature, intact and in contact with other cell types not present in the organoid system such as inflammatory cells and neurones.”

Referee #2:

Title: Single cell transcriptomics of human organoid-derived enteroendocrine cell populations from the small intestine

The authors have formulated a transcriptomic analysis of ileal and duodenal enteroendocrine cells (EECs) purified from human intestinal organoids on the basis of Venus fluorescent protein reporter expression downstream of the Chromogranin A promoter. EEC clusters at the transcript level are described according to hormone synthesis, and expression of nutrient-sensing and exocytotic machinery, and transcription factors. Expression patterns are mapped to a similar high impact study performed on NEUROG3 derived human EECs from organoids (Beumer *et al.*, 2020) and to EECs derived from the native intestine (Beumer *et al.*, 2020; Hickey *et al.*, 2023), where there is a high degree of congruence, and some notable differences. These findings are discussed in a sophisticated fashion.

There are two main limitations to the ms in its current form: (i) transcriptomic expression patterns are not corroborated by immunolabelling of EEC derived from the organoids; I think this would be an important step towards validating the organoids as a valuable model for functional analysis; (ii) functional analysis of exemplar 'nutrient-sensing'-'hormone secretion' transcriptional expression patterns identified in EEC-subtypes. These gaps in the current study of organoids are partially addressed by referring to published work. It is fair to say that the title states that the study is a transcriptional analysis. Nevertheless, these limitations should be discussed in the context of future directions. Comments on expected findings and their implications would be informative. Also, implications of the current findings for human health and disease would also be informative and provide potential impact.

Response: As this is a short focussed paper based on transcriptomics, we have addressed this in the discussion rather than by functional experiments which are covered in other studies.

Immunolabelling relies on verified antibodies, which for many nutrient sensing receptors are lacking. Studies using verified selective agonists or antagonists for a given receptor might also be employed

to verify functional expression (with the caveat of specificity) potentially giving information on functional relevance beyond “simple” translation of a transcript, but such studies in our view go beyond the focus of this short manuscript. However, we have added the following to the discussion:

“Supporting the relevance of the transcriptomic approach taken here, a number of studies have demonstrated that EECs in human organoids replicate physiological properties of the native enteroendocrine system, employing nutrient sensing machinery such as SGLT1, FFAR1 and GPBAR1 (Goldspink et al., 2020; Miedzybrodzka et al., 2021; Guccio, 2024). The ability to knock out candidate genes in organoids using CRISPR-Cas9 further offers a viable and valuable approach to validate the physiological relevance of individual genes identified in transcriptomic studies, complementing more traditional pharmacological approaches that are frequently hampered by limitations in drug specificity.”

Dear Dr Gribble,

Re: JP-RP-2024-287463R1 "Single cell transcriptomics of human organoid-derived enteroendocrine cell populations from the small intestine" by Christopher A. Smith, Van B Lu, Rula Bany Bakar, Emily Miedzybrodzka, Adam Davison, Deborah Goldspink, Frank Reimann, and Fiona Gribble

Thank you for submitting your manuscript to The Journal of Physiology. It has been assessed by a Reviewing Editor and by 0 expert referees and we are pleased to tell you that it is acceptable for publication following satisfactory revision.

REVISION CHECKLIST:

We look forward to receiving your revised submission.

Yours sincerely,

Kim Barrett
Senior Editor
The Journal of Physiology

EDITOR COMMENTS

Reviewing Editor:

Comments to the Author:

The authors have responded appropriately to the Reviewers comments. However, I note that a new paragraph referred to in the Response to Reviewers appears to have been omitted from the revised manuscript.

ie: "Although the data clustered broadly into cell types according to the dominantly expressed hormone, mirroring the classical single letter nomenclature, overlaps of hormone expression were apparent and are supported by many previous studies (Egerod et al., 2012; Habib et al., 2012; Haber et al., 2017; Kaelberer & Bohorquez, 2018) raising the question of whether it is time for the field to agree a new nomenclature for enteroendocrine cells that is both intuitive and representative. ."

Dear Kim and colleagues,

Thank you for noticing the mismatch between the final sentence in the manuscript and the version in the replies to referees. I have now corrected the final sentence in the manuscript (with many apologies for the error on my part), and also took the opportunity to update the reference Guccio et al, which now has a DOI (although still no volume and page numbers).

Best wishes

Fiona

Dear Professor Gribble,

Re: JP-RP-2024-287463R2 "Single cell transcriptomics of human organoid-derived enteroendocrine cell populations from the small intestine" by Christopher A. Smith, Van B Lu, Rula Bany Bakar, Emily Miedzybrodzka, Adam Davison, Deborah Goldspink, Frank Reimann, and Fiona Gribble

We are pleased to tell you that your paper has been accepted for publication in The Journal of Physiology.

Yours sincerely,

Kim Barrett
Senior Editor
The Journal of Physiology

If you would like to receive our 'Research Roundup', a monthly newsletter highlighting the cutting-edge research published in The Physiological Society's family of journals (The Journal of Physiology, Experimental Physiology, Physiological Reports, The Journal of Nutritional Physiology and The Journal of Precision Medicine: Health and Disease), please click this link, fill in your name and email address and select 'Research Roundup':

<https://www.physoc.org/journals-and-media/membernews>

- You can help your research get the attention it deserves! Check out Wiley's free Promotion Guide for best-practice recommendations for promoting your work at: www.wileyauthors.com/eo/guide. You can learn more about Wiley Editing Services which offers professional video, design, and writing services to create shareable video abstracts, infographics, conference posters, lay summaries, and research news stories for your research at: www.wileyauthors.com/eo/promotion.

The Corresponding Author will receive an email from Wiley with details on how to register or log-in to Wiley Authors Services where you will be able to place an order

EDITOR COMMENTS

Reviewing Editor:

Comments to the Author:
No further comments.